# Robotized Knee-Ankle-Foot Orthosis-Assisted Gait Training on Genu Recurvatum during Gait in Patients with Chronic Stroke: A Feasibility Study and Case Report

**DOI:** 10.3390/jcm12020415

**Published:** 2023-01-04

**Authors:** Yoko Takahashi, Kohsuke Okada, Tomoyuki Noda, Tatsuya Teramae, Takuya Nakamura, Koshiro Haruyama, Kohei Okuyama, Kengo Tsujimoto, Katsuhiro Mizuno, Jun Morimoto, Michiyuki Kawakami

**Affiliations:** 1Department of Physical Therapy, Faculty of Health Science, Juntendo University, Tokyo 113-8421, Japan; 2Department of Brain Robot Interface, Advanced Telecommunications Research Institute International, Kyoto 619-0288, Japan; 3Department of Rehabilitation Medicine, Keio University School of Medicine, Tokyo 160-8582, Japan; 4Department of Physical Rehabilitation, National Center Hospital, National Center of Neurology and Psychiatry, Tokyo 187-8551, Japan; 5Department of Rehabilitation Medicine, Tokai University School of Medicine, Kanagawa 259-1193, Japan; 6Graduate School of Informatics, Kyoto University, Kyoto 606-8501, Japan

**Keywords:** pneumatic artificial muscles, knee hyperextension, hemiplegia, lower extremity, walking, rehabilitation

## Abstract

Genu recurvatum (knee hyperextension) is a common problem after stroke. It is important to promote the coordination between knee and ankle movements during gait; however, no study has investigated how multi-joint assistance affects genu recurvatum. We are developing a gait training technique that uses robotized knee-ankle-foot orthosis (KAFO) to assists the knee and ankle joints simultaneously. This report aimed to investigate the safety of robotized KAFO-assisted gait training (Experiment 1) and a clinical trial to treat genu recurvatum in a patient with stroke (Experiment 2). Six healthy participants and eight patients with chronic stroke participated in Experiment 1. They received robotized KAFO-assisted gait training for one or 10 sessions. One patient with chronic stroke participated in Experiment 2 to investigate the effect of robotized KAFO-assisted gait training on genu recurvatum. The patient received the training for 30 min/day for nine days. The robot consisted of KAFO and an attached actuator of four pneumatic artificial muscles. The assistance parameters were adjusted by therapists to prevent genu recurvatum during gait. In Experiment 2, we evaluated the knee joint angle during overground gait, Fugl-Meyer Assessment of lower extremity (FMA-LE), modified Ashworth scale (MAS), Gait Assessment and Intervention Tool (G.A.I.T.), 10-m gait speed test, and 6-min walk test (6MWT) before and after the intervention without the robot. All participants completed the training in both experiments safely. In Experiment 2, genu recurvatum, FMA-LE, MAS, G.A.I.T., and 6MWT improved after robotized KAFO-assisted gait training. The results indicated that the multi-joint assistance robot may be effective for genu recurvatum after stroke.

## 1. Introduction

Genu recurvatum (knee hyperextension) refers to the extension of the knee in the affected leg beyond the neutral anatomical position during gait [1]. Genu recurvatum is a common disorder after stroke [2]. The incidence of patients with genu recurvatum due to hemiplegia has been reported to be 19.5–65% in those who could walk without assistance [3,4]. Problems associated with genu recurvatum include an increase in asymmetry and energy cost in walking [5], an increased impact due to foot contact in the knee extension position [6], and a risk of joint deformity due to pain and ligament extension [7]. Treatment for genu recurvatum is necessary for patients with frequent walking opportunities.

Previous studies listed potential causes of genu recurvatum, including weakness of the knee muscles, spasticity of knee extensors and ankle plantar flexors, production of excessive active tension with the ankle plantar flexors, forward pelvic tilt and lumbar hyperlordosis caused by weakness of the buttock muscles, and proprioceptive disorders [2,5,7]. The causes of genu recurvatum described in these previous studies included abnormal muscle activity of the hip, knee, or ankle joints. Indeed, the disorder is not only due to the knee joint, but also due to the coordination of multiple joints.

There are reports of genu recurvatum treatment in patients with stroke using various types of orthoses [8,9,10], functional electrical stimulation [11,12], electrogoniometric feedback [1], nerve block [13], and proprioceptive training using videographic observation [14]. However, reports of the after-effects of treatment are limited.

Recently, robot-assisted gait training has become widespread for patients with stroke, and numerous relevant meta-analyses and systematic reviews have been published. The devices used for training include the Lokomat (Hocoma AG, Zurich, Switzerland), Gait Trainer GT I/II (Reha-stim Medtec AG, Zurich, Switzerland), Anklebot (Interactive Motion Technologies, Watertown, MA, USA), and Walkbot (P&S Mechanics, Seoul, Republic of Korea); however, the Lokomat is widely used. Some recent reviews showed the effects of robot-assisted gait training on balance [15,16], gait speed [15,17], spatiotemporal gait parameters [18], walking independence [15,16,17,19,20], and endurance [19]. Despite several reported improvements in these quantitative parameters, studies that showed improved kinematics are limited [18,21]. Given this, Mao et al. [6] assessed the effects of a robotic knee orthosis on genu recurvatum in a patient with hemiplegia after brain tumor surgery. The robotic orthosis with an actuator on the knee joint only improved the degree of peak flexion and peak extension moment of the affected knee joint during gait but did not improve the degree of peak extension. Considering the reported lack of long-term effects of ankle movement assistance by functional electrical stimulation [11,12], it may be difficult to improve genu recurvatum using single-joint assistance. Thus, we hypothesized that a device that retrains the coordination of the knee and ankle joints may improve genu recurvatum.

Almost all exoskeletal robots use springs to assist the ankle joints. The robotized knee-ankle-foot orthosis (KAFO) is actuated by pneumatic artificial muscles (PAMs) [22]. This new device can independently modulate the assistance timing and power of the knee and ankle joints. Furthermore, PAMs can induce stronger power for ankle joint assistance compared to springs; hence, they may suppress genu recurvatum by inhibiting the strong ankle plantar flexion in the early stance phase.

It is therefore necessary to verify the safety and feasibility of this newly developed robotic device. We aimed to investigate (1) the feasibility of our robot-assisted gait training method in healthy individuals and patients with stroke, and (2) a case study to treat genu recurvatum in a patient with stroke.

## 2. Materials and Methods

This study consisted of two experiments: Experiment 1, verification of the safety and feasibility of our robotic device in healthy individuals and patients with chronic stroke; and Experiment 2, a single case study to investigate the effects of our robotic device in long-term intervention in a patient with chronic stroke.

The study protocol was approved by the local ethics committee (approval number: 20160053 and 20190246), and all tests were performed at Keio University School of Medicine. All participants provided written informed consent prior to enrolment. The study complied with the guidelines of the Declaration of Helsinki.

### 2.1. Participants

#### 2.1.1. Healthy Individuals

Six healthy individuals participated in Experiment 1. The mean age ± standard deviation of the participants was 31.8 ± 5.9 years. The inclusion criteria were healthy adults aged 20–80 years. None of the healthy individuals had a history of neurological disease or orthopedic disorders of the legs.

#### 2.1.2. Stroke Patients

We recruited nine patients with chronic stroke. The inclusion criteria were as follows: (1) patients with a stroke for more than six months; (2) patients with hemiparetic stroke and motor disability of the lower extremity; (3) first occurrence of stroke; (4) no medical history of fracture and injury by fall within three months preceding the study; (5) ability to understand and consent to the objectives and methods of the study; (6) ability to safely maintain a standing position, with or without assistance; and (7) presence of genu recurvatum during gait. The exclusion criteria were as follows: (1) serious cardiac disease; (2) uncontrolled hypertension; (3) acute systemic disease or fever; (4) recent pulmonary embolism, acute cor pulmonale, serious pulmonary hypertension, or recent pulmonary embolism; (5) serious liver or renal dysfunction; (6) serious orthopedic disease that bars exercise capability; (7) serious cognitive dysfunction or psychiatric disorder; (8) other metabolic abnormalities; (9) serious contractures of the joints of the legs; (10) implanted electronic pacing or defibrillation devices; (11) surgical history of shunting or clipping; and (12) medical history of epilepsy. Patients A–H participated in Experiment 1, and patient I participated in Experiment 2. All patients were community-ambulating, independently using ankle-foot orthosis (AFO). Some patients used a T-cane. The demographic and clinical characteristics of the study participants are summarized in Table 1.

### 2.2. Study Device

#### 2.2.1. Robotized Knee-Ankle-Foot Orthosis

Figure 1 illustrates the exoskeleton robotic device. The robotic device is the knee-ankle exoskeleton robot, which is an extension of the ankle exoskeleton robot, detailed in a prior report [22], to a knee-ankle exoskeleton robot. In the prior report [22], they first robotized the widely used AFO with a Klenzak double-channeled ankle joint instead of designing an ankle-joint exoskeleton robot from scratch, termed “the robotized AFO”. Thus, the robotic device in Figure 1 is termed “the robotized KAFO”. The robotized KAFO consists of the following four parts: the exoskeleton body consisting of the KAFO of metal struts, the actuator of the PAMs, the operation computers, and the control computer. As engineering researchers in our team aimed to develop a gait-assist robot that can be easily used in a clinical setting, they developed a novel robot, Modular Exoskeletal Joints, to actively drive a double-bar AFO. This modular joint is driven by the external PAMs to achieve both a large push-off force during the terminal stance phase and lightweight during the swing phase as same as the prior work [22]. Subsequently, the state of the art is that we applied this technology to KAFO and developed a robotized KAFO for patients with knee-ankle linkage problems in this study. The specifications of the robotized KAFO are listed in Table 2.

The Modular Exoskeletal Joints were attached to the knee and ankle joints of the KAFO to convert cable tension to joint torque. The forces generated by PAMs were transmitted to the Modular Exoskeletal Joints via the Bowden cables. Stator of the Modular Exoskeletal Joint is attached to upper link of the orthosis, and the mover of the Modular Exoskeletal Joint is attached to lower link of the orthosis. The robot assists were provided by the Modular Exoskeletal Joints driving the knee and ankle joints of the orthosis.

The actuators were four nested chamber PAMs (NcPAMs) attached to the two of exoskeletal module joint of the knee and ankle joints. One exoskeletal modular joint is actuated by an antagonistic pair of NcPAMs. The NcPAMs can be located away from the exoskeleton, thanks to Bowden cables transmission system. The NcPAM unit was attached to the back of the participants in a state of suspension from the unloading device to make it feel light, although the NcPAMs are light weight. Therefore, the robotized KAFO weight borne by the participant was 2.9 kg. The KAFO is an exoskeleton that is not mechanically strong enough to directly receive the large radial force generated by the NcPAMs; therefore, it was necessary to design an exoskeletal joint that could only transmit the torque to the knee and ankle joints while supporting the large radial force from the exoskeletal joint structure. This structure also makes it possible to reduce the weight of the robotic exoskeleton. The functions of the two NcPAMs connected to the knee joint were knee extension and flexion, and those of the two NcPAMs connected to the ankle joint were plantarflexion and dorsiflexion. The robotized KAFO joint can be inherently compliant when NcPAMs are activated. More importantly, the joint also can be mechanically transparent and backdrivable, e.g., when the NcPAMs are not activated, the knee joint and ankle joint can be free thanks to NcPAMs without any feedback control.

#### 2.2.2. Assist Setting

The robotized KAFO is an assistive device that can allow a patient to move his or her joints by themselves. It also inhibits abnormal joint motion and assists in poor joint motion. The timing and amount of assistance can be changed to suit each patient; therefore, it responds flexibly to patients with various gait peculiarities. We regulated the assistive power and timing of each NcPAM using a parameter control device. The adjustment of assistive timing was applied as the feed-forward regulation based on the identification of the walking phase. The walking phase was identified by an algorithm based on foot pressure data obtained from both foot force-sensing register (FSR) sensors. The FSR sensors were placed on the balls of the great toes and heels. The FSR value during unassisted walking was used to identify the walking phase of each patient. Normalized data were linearized, and the gait cycle was identified based on the linearized values. The timing to begin and end the assist were independently adjusted in four motion directions with a value between 1% and 100%, with one gait cycle being 100%.

The assistive parameter was adjusted to prevent genu recurvatum, enhance push-off during the terminal stance phase, and achieve foot clearance and heel contact during the swing phase. Genu recurvatum was prevented by adjusting the parameter to prevent the knee joint angle from exceeding 0°. We evaluated the foot pressure of each FSR sensor, and the joint angle of the knee and ankle of the robot using a monitor through robot-assisted gait training and adjusted the assistive parameter corresponding to changes in gait. In Experiment 2, we introduced a real-time evaluation system for the knee angle of a patient to achieve a higher effect on genu recurvatum. An electric goniometer attached to the patient’s paralyzed knee joint was synchronized with the robot to adjust the assistive parameters according to the patient’s actual knee-joint angle.

### 2.3. Experimental Procedure

#### 2.3.1. Experiment 1

We conducted Experiment 1 to confirm the safety of robotized KAFO-assisted gait training for the intervention because this trial is the first opportunity that the robotized KAFO was clinically used. First, we developed a safety assessment (Appendix A) to forestall the possible the risks of performing robot-assisted gait training through experiments in six healthy participants. Experienced doctors, physical therapists, and robot developers participated in safety assessment development. The safety assessment consisted of the following three parts: (A) at the time of robot wearing (physical condition, risk assessment of robot and harness wearing, and risk assessment of environment); (B) during training (presence of unexpected assistance, presence of pain due to orthosis and leg contact/excessive leg motion/antagonism between robot assist and active muscle movement, and physical condition); and (C) after taking off the robot (skin condition at the contact point of the robot or harness, physical condition, and state of gait). The assessment included items about the participant’s physical condition or presence of pain and items about the state of gait to check for risks, such as instability or falls due to change in gait caused by training.

Second, we assessed the safety of both single-day and multi-day interventions in patients with stroke. Four patients with stroke (ID A, B, C, D) received single-day robotized KAFO-assisted gait training on a treadmill (20 min in total). Another four patients with stroke (ID E, F, G, H) received robotized KAFO-assisted gait training for 10 days. The participants wore the robot with the assistance of physical therapists for 5 min. The training was carried out by attaching a harness connected to a body-weight support device to prevent falls, but the setting of the body-weight support was 0 kg.

#### 2.3.2. Experiment 2

Since the safety of robotized KAFO-assisted gait training and the feasibility of multiday interventions were confirmed in Experiment 1, Experiment 2 was conducted to preliminarily examine the therapeutic effect of genu recurvatum. A patient with stroke (patient I) received robotized KAFO-assisted gait training on a treadmill for 30 min/day, four days a week, for nine days in total. Patient I received no other physical therapy during robotized KAFO-assisted gait training and at least 1 month prior to the intervention. A session consisted of the period of assistive parameter adjustment for 6 min and three trials of intervention for 8 min, for a total of 30 min. Patient I wore the robot with assistance from physical therapists within 5 min. The training was carried out by attaching a harness connected to the body-weight support device for fall prevention, but the setting of the body-weight support was 0 kg. During training, we instructed the patient to walk with the image of relearning the movement of the leg in accordance with the robot assistance. Real-time feedback of the knee joint angle from an electronic goniometer was also applied. After training, a physical therapist instructed the patient on overground gait for 5 min to prevent falls due to changes in gait pattern. We performed assessments before (pre) and after nine days of training (post). Pre-assessments were done the day before the intervention start date, and post-assessments were done the day after the intervention end. We also evaluated the safety of the training using our safety assessment.

### 2.4. Assessments

#### 2.4.1. Kinematic Data during Overground Gait

We used a three-dimensional (3D) motion capture system (Vicon, Vicon Motion Systems. Ltd., Oxford, UK) for gait analysis. The reflective marker sets were chosen according to the plug-in gait lower-body model. Sixteen markers were attached to anatomical landmarks on both sides as follows: anterior superior iliac, posterior superior iliac, thigh, knee, tibia, ankle, toe, and heel. We sampled motion data at a frequency of 100 Hz. Foot sensors were also used to identify the gait cycles. A total of 12 gait cycles were measured.

During gait analysis, we also collected electromyography (EMG) data from eight muscles on the affected side, namely, the tibialis anterior muscle (TA), medial gastrocnemius muscle (MG), soleus muscle (SOL), rectus femoris muscle (RF), vastus medialis muscle (VM), semitendinosus muscle (ST), biceps femoris muscle (BF), and gluteus maximus muscle (GM). We used a wireless surface EMG system (Telemyo, Noraxon U.S.A. Inc., Arizona, USA) and sampled EMG data at 1500 Hz.

#### 2.4.2. Clinical Assessments

Clinical assessments included the Fugl-Meyer Assessment of lower extremity [23] and modified Ashworth scale (MAS) [24]. Gait ability assessments included a 10-m gait speed test, a 6-min walk test (6MWT), and the Gait Assessment and Intervention Tool (G.A.I.T.) [25].

The FMA is a commonly used assessment to rate the recovery of motor function. We used MAS to assess spasticity in the quadriceps, hamstrings, and triceps surae muscles. This scale includes grades from 0 to 4, a high score indicates severe spasticity. A 10-m gait speed test was performed at both normal and fast speeds. The 6MWT was assessed according to the American thoracic society guidelines [26]. We wanted to reproduce the patient’s usual walking at both inside and outside of her house; therefore, the 6MWT was assessed in two situations. First, the patient walked barefoot and without a cane, just as she walked inside her home. Afterwards, the patient wore an AFO and walked using a T-cane, just as she walked outdoors. Regarding observational assessment of gait patterns, we used the G.A.I.T., which was developed to evaluate coordinated gait components after neural injury. The examiner rated the gait components based on the video of the patient’s walking. The total score was 62, and a lower score was preferred.

#### 2.4.3. Data Analysis of Kinematic Data

We analyzed 3D and EMG data using the MATLAB software (MathWorks, Natick, MA, USA). Both sets of data were time-normalized for each gait cycle. The mean value of the joint angle of the total gait cycle was calculated for each time points (0–100%, total 101 points) and plotted on the graph. EMG data were filtered at 20–450 Hz using a band-pass filter (4th Butterworth filter). We calculated the root mean square (RMS) with a time window of 200 ms and convolution interval of 1/1500 s. The 101 points RMS for 1 gait cycle were calculated and averaged for 12 steps. This averaged RMS was normalized with the mean RMS of 101 points. This ‘normalized EMG’ was plotted to the graph as representative data. Foot pressure of the heel and ball of the foot were time-normalized for each stance phase and calculated as the average of 12 steps.

## 3. Results

### 3.1. Safety of Robot-Assisted Gait Training

In both Experiments, all participants safely completed the training. All participants showed no considerable change in physical condition, had no problems with robot fitting, no pain during training, and no dangerous change in the state of gait. Four patients with stroke completed single-day intervention (20 min in total). Another four patients with stroke completed 20 min/day interventions for 10 days in total. One patient showed contact of the fibular head with the brace by the safety assessment before the training (robot fitting section). Since the contact area was padded before the training, the patient could complete training without any pain or adverse events. There was also no trouble continuing the intervention for multiple days. It was confirmed that robotized KAFO-assisted gait training can be performed safely by using the safety assessment, even with a multiday intervention, so we moved on to Experiment 2.

In Experiment 2, slight pain in the lateral malleolus was noted on the affected side by the safety assessment before the training (robot fitting section) on the first intervention day. The cause of the pain was identified as contact between the shoe and the medial malleolus, so the contact area was padded. Subject then completed the intervention without pain.

### 3.2. Experiment 2

The results of Experiment 2 are presented in Table 3. Patient I showed improvement in the peak knee extension angle of the affected side during the stance phase of overground gait without the robotic device (pre: −13.7°; post: 12.1°) (Figure 2). The patient also showed improvement in the peak knee flexion angle of the affected side during the swing phase (pre: 12.5°, post: 33.3°). That is, after the training period, genu recurvatum did not appear during overground gait even without robotic assistance. The peak ankle plantarflexion angle changed from −34.6° to −31.1°, and the peak ankle dorsiflexion angle changed from 2.7° to 15.0° (Figure 2). The EMG data are shown in Figure 3. We compared normalized RMS before and after training, and the activation patterns of the RF and VM on the affected side during the early stance phase (immediately after heel contact) changed after training. The VM in the early stance phase was especially activated after training. The GM activity pattern during the stance phase also changed after training. The foot pressure data are shown in Figure 4. Heel pressure responded throughout the stance phase before training (Figure 4A), however, after training the response decreased in about 80% of the stance phase (Figure 4B). Ball of the foot pressure was slow to rise in response before training (Figure 4C), and the reaction started in the latter half of the stance phase, after training, the reaction started in about 20% of the stance phase (Figure 4D). Data from the two sensors suggested an increase in forward load transfer in the stance phase.

The FMA-LE improved compared to the pre data (pre: 20, post: 25). The sub-items that showed improvement were as follows: E-I: reflex activities of the flexor and extensor reflexes could be elicited (pre: 2, post 4), E-IIa: volitional movement could be performed within the dynamic flexor synergy (pre: 4, post: 5), and E-V: normal reflex activity (pre: 1, post 2) was noted. The MAS of the quadriceps (pre: 2; post: 0), hamstrings (pre: 1; post: 0), and triceps surae (pre: 1+, post: 1) improved.

The distance covered in the 6MWT also improved from 266 m to 277 m while barefoot and without a cane. G.A.I.T. also showed improvement (pre: 36, post: 32).

## 4. Discussion

We found that robotized KAFO-assisted gait training could be performed safely using the safety assessment to prevent risk. In patients with chronic stroke, long-term robotized KAFO-assisted gait training may improve genu recurvatum during overground gait without the use of a robotic device, and motor function of the paralyzed lower limb. Changes in the pattern of muscle activities during overground gait were also observed with the improvement of genu recurvatum. We consider that robotized KAFO-assisted gait training has a therapeutic effect on genu recurvatum in patients with stroke.

### 4.1. Safety of Robotized KAFO-Assisted Gait Training

Robotized KAFO-assisted gait training was safely performed in all participants. We were also able to construct the safety assessment of robot-assisted gait training by conducting Experiment 1. Previous studies have assessed the safety of robot-assisted gait training in patients with neurological disorders using a self-report questionnaire [27], assessment of pain [28], recording of adverse events [27,28,29,30], and assessment using the U.S. Food and Drug Administration’s list of known and unforeseen adverse events [31]. Before the training, we listed the possible risks and adverse events and included the safety assessment. As a result, we were able to detect the pain caused by fitting the robot in order to prevent adverse events. Using safety assessment may make it possible to perform robot-assisted gait training safely. It is necessary to analyze many differences in robotic devices and diseases; however, we suggest that this safety assessment is potentially useful for various robotic devices and robot-assisted gait rehabilitation.

### 4.2. Case Study of 9 Days Intervention on Genu Recurvatum (Experiment 2)

The duration between the onset of stroke in patient I and her inclusion in this study was 3.7 years. Genu recurvatum is habitual; therefore, the patient had difficulty in preventing genu recurvatum during gait by herself before the intervention. However, nine days after the intervention, the patient achieved overground gait without genu recurvatum, even without the use of the robotized KAFO. Robotized KAFO-assisted training can stimulate the learning of joint movements while preventing genu recurvatum.

In a previous report on robotic intervention for genu recurvatum, a 15-day intervention that involved the use of a robotized knee orthosis could not prevent the appearance of knee hyperextension during both the stance and the swing phases [6]. The causes of genu recurvatum are not only due to disorders in the knee joint, but also due to movement disorders of the hip and ankle joints [2,5,7]. Therefore, gait training may be more effective if devices that assist both the knee and ankle joints are used during training sessions. A previous study examined kinematic changes before and after robot-assisted gait training in patients with chronic stroke; however, the study did not focus solely on genu recurvatum [21]. They found that Lokomat-assisted gait training for four weeks did not change the average coefficient of correspondence and the peak joint angle of flexion/extension of the hip and knee. Although Lokomat is an exoskeleton-type gait assist robot, changing the kinematics of a stroke patient’s gait could be relatively challenging. Our robotized KAFO is also an exoskeleton type, but any link of the robotized KAFO is not fixed while root link of the Lokomat is fixed to stational flame. One of its novel features is that the assist settings can be changed in detail for each patient. Furthermore, it may be that it is important to change the assist setting in order to change the kinematics of the gait of a patient with stroke with high individuality.

The patient walked barefoot overground before the intervention and this resulted in less angular changes between the knee and ankle on the affected side and loss of coordination between the knee and ankle joints. The training focused on heel contact with the knee in a slightly flexed position, promotion of lower leg forward tilting after heel contact, and relearning of push-off in the pre-swing phase. The device can independently adjust the timing and power of the assist in each joint motion, and it may have enabled tailor-made assistance for her gait pattern. Furthermore, real-time feedback of the knee joint angle was also performed using a monitor so that the patient could be aware of heel contact with the knee in a slightly flexed position. As a result, the patient acquired an overground gait without genu recurvatum. We suggest that relearning joint motion with robot assistance and motor learning with angle feedback may improve genu recurvatum. Considering that proprioceptive training using videographic observation improved genu recurvatum [14] and electrogoniometric feedback of the knee joint showed constant effects on genu recurvatum [2,32,33], motor learning with joint angle feedback also plays an important role in the improvement of genu recurvatum.

In Experiment 2, the FMA-LE also showed improvement in E-I, E-II, E-IV, and F. Previous studies in patients with acute and sub-acute stroke have reported improvement in FMA-LE with robot-assisted gait training [34,35,36,37,38,39]. Although some studies have shown similar improvements in FMA-LE after robot-assisted gait training in patients with chronic stroke [40,41], the number of sessions or duration of intervention were longer than those used in our intervention. The robotized KAFO can independently adjust the knee flexion/extension assistance and ankle plantar flexion/dorsiflexion assistance, as required. Furthermore, the MAS scores of the quadriceps, hamstrings, and triceps surae muscles improved after this intervention. The causes of spasticity include neural changes, muscle atrophy, and muscle contractures [41,42,43,44]. Increased motion of the knee joint during gait may have resulted in stretching and shortening of the thigh muscles and could have affected the viscoelasticity of the muscle-tendon complex. Spasticity in the quadriceps or triceps surae muscle is thought to be part of the cause of genu recurvatum [2,5,7]. Reduced spasticity of these muscles may have led to reduced extensor synergy and improved genu recurvatum.

In addition to the qualitative component of lower-limb movement, it is important not to lower the efficiency of gait. The walking speed after training did not get worse, while controlling for genu recurvatum. Rather, the results of 6MWT with barefoot showed improved walking efficiency over long distances. After the intervention, as assessed using the sub-items of the FMA-LE (E-II and E-IV), flexor synergy and voluntary ankle dorsiflexion improved. Furthermore, the EMG activation pattern of the knee extensors changed. It has also been reported that the activity of the knee extensor muscle increases immediately after initial foot contact (in the early stance phase) in the gait of healthy individuals [3]. Before the intervention, the patient showed low VM activity in the early stance phase due to genu recurvatum. After the intervention, VM activity increased at the same time as heel contact did. Simultaneously, GM activity decreased. It is generally known that the GM is active in maintaining forward movement of the center of gravity against the impact of the ground at the initial contact [45]. The decrease in GM activity at initial contact in this patient may be related to the reduced impact of contact with the ground owing to regaining contact with a slightly flexed knee position. In relation to these changes in muscle activities and kinematics, foot pressure data also suggested increased load transfer to forward in the affected side during the stance phase. Since this was a single case study, the training effect of treadmill walking should be considered. There are limited reports showing that treadmill walking training alone changed the kinematics of the affected lower limb joint in patients with stroke [46]. The change that Druzbicki et al. [46] reported in hemiparetic knee angle before and after training was about 5° in the mean value of 10 patients. In comparison with this result, our case showed a larger knee and ankle joint angle change. The changes in leg motor function, and muscle activation pattern with kinematics change on the affected side may contribute to improved walking efficacy. Further studies are needed to investigate muscle activity after the training.

### 4.3. Clinical Implication

Many patients suffer from genu recurvatum after a stroke [3,4] and show decreased walking efficacy or pain; therefore, treatment for genu recurvatum is required in this patient population. Our robotic device has the potential to be therapeutic device for genu recurvatum.

### 4.4. Limitations

As this study was conducted with a very small number of participants/patients. Statistical analysis was also not performed; therefore, the extent to which the effects of the intervention could be verified is limited. Follow-up assessments after completion the intervention are unavailable and will be needed in the future. Another limitation of this study was that we could not assess the activity of daily living of the participants. Additionally, robotized KAFO-assisted gait training was performed on a treadmill, and in Experiment 2, joint angle feedback was also performed; therefore, there was no control group. A randomized controlled trial with a sufficient sample size is necessary in the future.

## Figures and Tables

**Figure 1 jcm-12-00415-f001:**
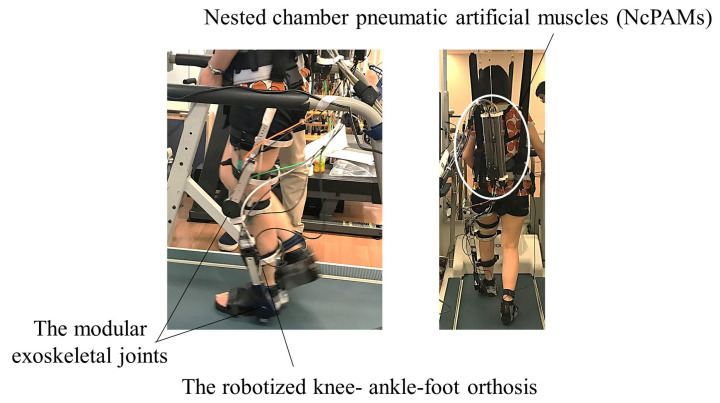
Robotized knee-ankle-foot orthosis. We used a robotized knee-ankle-foot orthosis. The device consisted of an orthosis, Modular Exoskeletal Joints, nested chamber pneumatic artificial muscles (NcPAMs), and a control personal computer. The unit of NcPAMs was carried on the patient’s back and was relieved by a body-weight support device. The four PAMs assisted in knee flexion and extension, ankle dorsiflexion, and plantar flexion.

**Figure 2 jcm-12-00415-f002:**
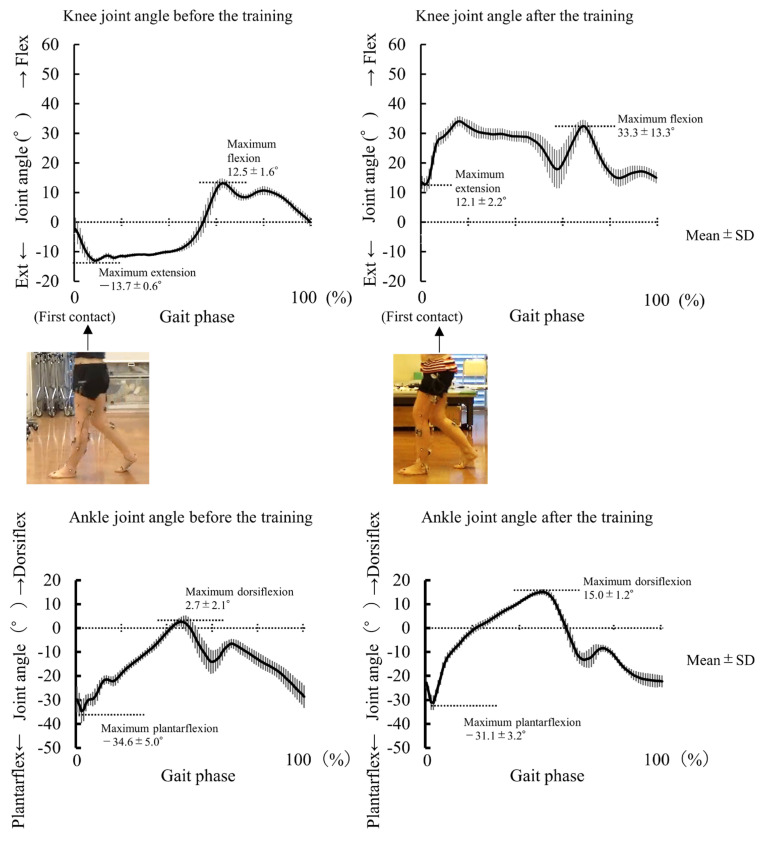
Knee and ankle joints angle on the affected side during gait before and after nine days of robot-assisted gait. The graphs indicate the knee and ankle joints angle during barefoot overground walking before (**left**) and after training (**right**). The horizontal axis indicates one gait cycle, with 0% indicating initial foot contact on the affected side and 100% indicating the next contact. The thick solid line represents the mean angle, and the thin vertical line represents the standard deviation. As the patient’s photographs show, genu recurvatum was seen before the training, but it disappeared after the training. The maximum knee extension angle changed from −13.7° to 12.1°. The maximum knee flexion angle changed from 12.5° to 33.3°. The maximum ankle plantarflexion angle changed from −34.6° to −31.1°. The maximum ankle dorsiflexion angle changed from 2.7° to 15.0°.

**Figure 3 jcm-12-00415-f003:**
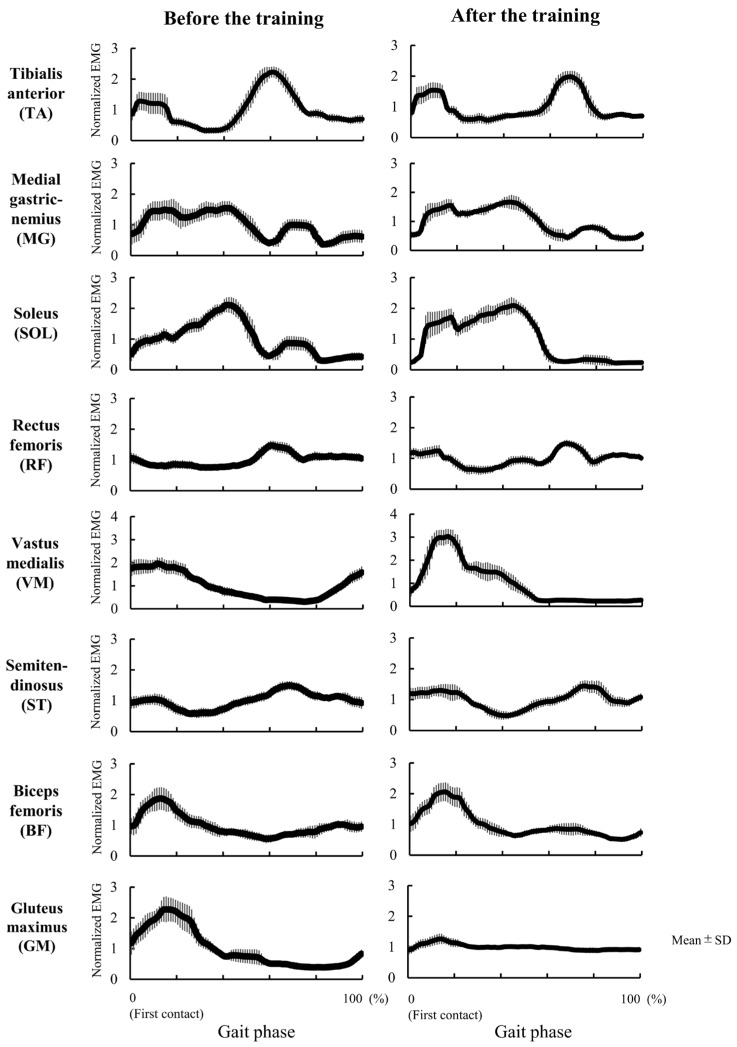
Electromyography (EMG) on the affected side during gait before and after nine days of robot-assisted gait training. Graphs indicate normalized electromyography (EMG) during barefoot overground walking before (**left**) and after training (**right**). The horizontal axis indicates one gait cycle, with 0% indicating initial foot contact on the affected side and 100% indicating the next contact. The thick solid line represents the mean normalized EMG, and the thin vertical line represents the standard deviation. We assessed eight muscles: the tibialis anterior muscle, the medial gastrocnemius muscle, the soleus muscle, the rectus femoris muscle, the vastus medialis muscle, the semitendinosus muscle, the biceps femoris muscle, and the gluteus maximus muscle. After the training, the genu recurvatum disappeared, and the activity of the vastus medialis muscle increased in the early stance phase.

**Figure 4 jcm-12-00415-f004:**
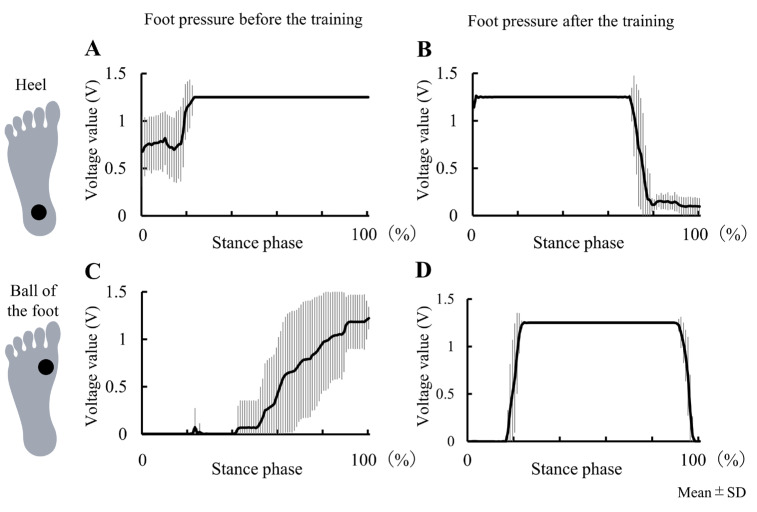
Foot pressures of the heel and the ball of the foot on the affected side during gait (stance phase) before and after nine days of robot-assisted gait training. Graphs indicate the voltage values of the foot pressures of the heel and the ball of the foot on the affected side. The horizontal axis indicates one stance phase, with 0% indicating initial foot contact on the affected side and 100% indicating toe off on the affected side. The thick solid line represents the mean voltage value, and the thin vertical line represents the standard deviation. Heel pressure responded throughout the stance phase before training (**A**), however, after training the response decreased in about 80% of the stance phase (**B**). Ball of the foot pressure was slow to rise in response before training (**C**), and the reaction started in the latter half of the stance phase, after training, the reaction started in about 20% of the stance phase (**D**). Data from the two sensors suggested an increase in forward load transfer in the stance phase.

**Table 1 jcm-12-00415-t001:** Patients’ characteristics.

ID	Diagnosis	Age (Years)	Sex	Affected Side	TFO (Month)	FMA-LE	Modified Ashworth Scale			Experiment
							Quad	Hamst	TS	
A	Cerebral hemorrhage	50	Female	Right	40	19	0	0	1+	1
B	Cerebral infarction	21	Male	Right	47	25	0	0	2	1
C	Cerebral hemorrhage	18	Female	Left	12	23	1	1	1+	1
D	Cerebral infarction	64	Male	Left	24	20	1+	0	2	1
E	Cerebral hemorrhage	53	Female	Left	30	29	0	1+	1	1
F	Cerebral hemorrhage	30	Female	Right	19	20	0	0	2	1
G	Cerebral infarction	68	Male	Right	15	19	0	0	1	1
H	Cerebral hemorrhage	64	Male	Right	88	18	1+	0	3	1
I	Cerebral infarction	47	Female	Left	44	20	2	1	1+	2

TFO, time from stroke onset; FMA-LE, Fugl-Meyer Assessment of lower extremity; Quad, quadriceps muscle; Hamst, hamstrings muscle; TS, triceps surae muscle.

**Table 2 jcm-12-00415-t002:** The specification of the robotized knee-ankle-foot orthosis.

Spec	Value
Weight	
-Total	5.0 kg
-Nested chamber pneumatic artificial muscles (NcPAMs) unit	2.1 kg
-Exoskeleton and modular exoskeletal joint	2.9 kg
Encoder	Optical 3 phase 16,000 counts per revolution
Max pressure	0.8 MPa
Max peak torque	48.0 Nm
NcPAM diameter	20 mm each

**Table 3 jcm-12-00415-t003:** Results of clinical assessments in Experiment 2 (Patient I).

Clinical Assessments		Pre	Post
Fugl-Meyer Assessment of lower extremity		20	25
(Full score 34 points)			
(E-I/E-IIa/E-IIb/E-III/E-IV/E-V/F)		(2/4/7/2/1/0/4)	(4/5/7/2/2/0/5)
Modified Ashworth scale	Quadriceps muscle	2	0
	Hamstrings muscle	1	0
	Triceps surae muscle	1+	1
10-m gait speed test	Normal		
	Time (s)	14.7	14.5
	Steps (steps)	21	21
	Cadence (steps/min)	85.7	86.9
	Fast		
	Time (s)	12.9	12.7
	Steps (steps)	21	20
	Cadence (steps/min)	97.7	94.5
6-min walk test	Barefoot, no cane (m)	266	277
	With ankle-foot orthosis, T-cane (m)	345	343
Gait Assessment and Intervention Tool		36	32

## Data Availability

The data that support the findings of this study are available from the corresponding author; however, restrictions apply to the availability of these data, which were used under license for the current study, and hence, are not publicly available. However, the data are available from the authors upon reasonable request and with permission from the corresponding author.

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
