# Peer review of "Robotized Knee-Ankle-Foot Orthosis-Assisted Gait Training on Genu Recurvatum during Gait in Patients with Chronic Stroke: A Feasibility Study and Case Report"

_jcm, 2023, doi:10.3390/jcm12020415_

Round 1

Reviewer 1 Report

In the presented study Authors assessed the safety of a robotized knee-ankle-foot-orthosis for use during an assisted gait training routine. For this, a safety assessment document is developed and presented. Furthermore, a case study is reported with preliminary results on the effectiveness of the device for treatment of genu recurvatum caused by chronic stroke. Although the limited results, the manuscript can be considered in line with the proposed article type, i.e., brief report. However, contents could be further improved, as well as, some points need to be addressed.

Comments are listed below:

In order to avoid repetitions, e.g., about PAMs description or device specifications, I strongly suggest to merge sections 2.2.1. with 2.2.2.

Line 171-174: if not already reported in previous studies by Authors, It would be more appropriate to include in the present manuscript all available information about used materials and methods. In case, as supplementary materials. Otherwise, in my opinion, reproducibility of the study is not guaranteed.

Regarding results of the study, it would be interesting to report variations related to the foot pressures of the affected side. Are these data taken into account? Moreover, as reported in Figure 2 for the knee, please, include data about the ankle joint angle and, if the case, discuss the corresponding findings.

For sake of readability, I would suggest to remove the 3.3 “Figure and Tables” subheading and to move the description of the figures within the corresponding captions.

Line: 341-343: the literature (please, see below) reports several other studies involving multiarticular exoskeleton robots, applied to the lower limb, which actively assist two or more joints. Please, better argue your statement. Otherwise, in my opinion, this statement should be omitted.

References:

D. Huamanchahua, C. L. Otarola-Ruiz, A. Quispe-Piña and E. J. De La Torre-Velarde, "Knee and Ankle Exoskeletons for Motor Rehabilitation: A Technology Review," 2022 IEEE International IOT, Electronics and Mechatronics Conference (IEMTRONICS), 2022, pp. 1-7, doi: 10.1109/IEMTRONICS55184.2022.9795725

Rodríguez-Fernández, A., Lobo-Prat, J. & Font-Llagunes, J.M. Systematic review on wearable lower-limb exoskeletons for gait training in neuromuscular impairments. J NeuroEngineering Rehabil 18, 22 (2021). https://doi.org/10.1186/s12984-021-00815-5

Reviewer 2 Report

I would like to thank the authors very much for addressing this interesting topic. Any work that strives to find more effective rehabilitation, especially one assessed by objective forms of feedback (like EMG), is a step toward the evidence-based rehabilitation so desperately needed.

Major issues:
1. My biggest concern, however, is the very small group of people who took part in the experiment, especially in its second part. What is the reason for such a small number of people included in the experiment? And why only one person in the second part of the experiment? Please complete this clarification and consider adding it to the work limitation.

2. What were the conditions for selecting healthy people for the experiment and if there were any, or were they just 6 random people (without neurological or orthopedic diseases)? In the study, the description is only 3 lines. What about other conditions such as fitness level, body weight, height? With such a small number of patients (only 3 people) I actually don't quite understand the point of including healthy people here. I would like to ask the authors to clarify.

3. Patients vary strongly in age. Two of them are about 50 years old, and the third is half that age. Did they also differ significantly in their level of physical fitness? What is the reason for dividing the patients into Experiment 1 and Experiment 2? And why did all three not participate in part 1 and part 2? I miss a more detailed explanation in the study.

4. You have written in line 207 that the first part of the experiment is to evaluate "the safety of robotized KAFO-assisted gait training for long-term intervention". I don't quite agree that one 20-minute workout can in any way evaluate the safety of training as a long-term intervention. We still don't know what would happen after multiple training sessions. I think the second part of the experiment gives us more information on this. However, I would suggest removing that phrase "long-term". One 20-minute training session tells us whether the presented KAFO system is safe during training, but says nothing about long-term safety profile.

5. Please tell more precisely when the pre- and post-training tests were done, and please complete this description of the 2.5 assessments - was it immediately before and after the training (on the same day), or were there any breaks?

6. You use the phrase "normalized EMG" (e.g., in Figure 3). Please elaborate and add in the text what "normalized EMG" is. 

7. After all, how do we know that the improvement in the performance of the patient from Experiment 2 is not simply the result of intensive (10-day) trials of training to walk correctly, or additional activities, rather than the equipment itself? This is just one person. 

8. The title of the paper suggests a study. However, please consider including that it is more of a case report.

Minor issues:

1. Lines 277-279 - you did not remove the instructions from the template from the editors of MDPI.

2. Line 328 - "Patient C completed 10 days robotized knee-ankle-foot orthosis" - you seem to broke off in mid-sentence. The end of the sentence is missing here.

Thank you for your efforts in creating this work, and I look forward to reading future results on a larger group of patients and with longer follow-up.

Round 2

Reviewer 2 Report

Thank you to the authors for taking into account all my comments. The work has definitely gained some value.